# Statistical Analysis of Magnetopause Crossings at Lunar Distances

Johannes Z. D. Mieth[1], Dennis Frühauff[1], and Karl-Heinz Glassmeier[1]

[1]Technische Universität Braunschweig, Institut für Geophysik und extraterrestrische Physik, Mendelsohnstraße 3, 38106 Braunschweig, Germany

**Correspondence:** Johannes Z. D. Mieth (j.mieth@tu-braunschweig.de)

**Abstract.** Different magnetopause models with a diverse level of complexity are in use. They have in common to be mainly based on near-earth observations, i.e., they use measurements at distances of about $\pm 10$ Earth radii along the GSM $x$-axis. Only very few observations of magnetopause crossings at larger distances are used for model fitting. In this study we compare position and normal direction predictions of the Shue et al. (1997) magnetopause model with actual observations of magne-

topause crossings identified using the ARTEMIS spacecraft at lunar distance, about 60 Earth radii. We find differences in the location prediction between model and actual observation but good agreement in predictions about the magnetopause normal direction.

## 1   Introduction

The magnetopause plays an important role for space weather processes as it is the primary interaction zone between the

solar wind plasma and the Earth's magnetosphere. The magnetopause is defined as the boundary between solar wind and magnetospheric plasma which can not be penetrated by the solar wind (Baumjohann and Treumann, 1996). In case of an equilibrium magnetopause this is the plane where the solar wind pressure is balanced by the Earth's magnetic field pressure (e.g. Glassmeier et al., 2008). In 1997 Shue and co-workers presented a very simple model to predict the magnetopause (MP) position under different solar wind (SW) conditions (Shue et al., 1997). Additionally to the location prediction it is possible

to also deduce the MP normal direction. Using data of magnetopause crossings of the ISEE 1 and 2, AMPTE/IRM, and IMP 8 satellites they modelled the magnetopause radial distance $r$ with the functional form $r = r_0 \left[ 2/\left(1 + \cos\theta\right)\right]^{\alpha}$. Here $r_0$, $\theta$, and $\alpha$ denote the standoff distance, the angle between the Sun-Earth line and the direction of $r$, and the magnetopause flaring parameter, respectively (Fig.3). Shue et al. (1997) modelled the MP location to be only depended on the $B_z$ component of the interplanetary magnetic field (IMF) and the solar wind dynamic pressure $D_p$. This functional model is mathematical

axially symmetric around the $x$-axis in solar wind aberration corrected geocentric solar ecliptic (GSE) and geocentric solar magnetospheric (GSM) coordinates (Hapgood, 1992). Measurements used for the determination of the fitting parameters are mainly from distances of $\pm 10$ Earth radii ($R_{\mathrm{E}}$) on the $x$-axis with only a few data points expanding up to about $30\,R_{\mathrm{E}}$ downtail. As precise the Shue model is, it requires further observations from higher latitudes as well as crossings further downtail from the Earth to provide a more realistic 3D magnetopause model (Shue and Song, 2002). Extensions of the Shue model were

thus presented by, e.g., Lin et al. (2010) and Wang et al. (2013). However, all the proposed models are still characterized by using only a very limited number of measurements at greater distances downtail. This is where our study contributes. By using

plasma and magnetic field measurements from the ARTEMIS mission we validate the Shue model at radial distances of about $60\,\mathrm{R_E}$ downtail.

## 2    Data selection and analysis procedure

The *Acceleration, Reconnection, Turbulence, and Electrodynamics of the Moon's Interaction with the Sun* (ARTEMIS) Mission (Angelopoulos, 2010) provides long term measurements of the plasma environment in the terrestrial magnetosphere at lunar distances, about $60\,\mathrm{R_E}$. Since 2011 the two spacecraft THB and THC orbit the Moon and provide excellent measurements of the plasma environment there. The THB and THC spacecraft originate from the THEMIS mission (Angelopoulos, 2008), a NASA Medium-Class Explorers (MIDEX) mission, launched on February 17, 2007 and designed to investigate the trigger mechanisms and evolution of magnetospheric substorms. Five identical spacecraft were put into Earth's orbit to line up along the magnetotail. After the primary mission phase the two outermost spacecraft were lifted into a lunar orbit. Since May 2011 both probes are in stable equatorial and eccentrical orbits.

### 2.1    Observations

Our study covers a time span of five years, starting January 2011 and lasting until December 2015. Different types of data products are used to determine magnetopause position and direction. The electrostatic analyzer (ESA) (McFadden et al., 2008) provides ion and electron flux density over a broad energy band from only a few $\mathrm{eV}$ up to $30\,\mathrm{keV}$. We use time resolved ion energy data with a resolution of about $3\,\mathrm{s}$ in this study. In order to generate this data set, measurements with higher temporal resolution are integrated over a spin period of the spacecraft. The plasma data are complemented by measurements from the ARTEMIS fluxgate magnetometer (FGM) (Auster et al., 2008), providing vector magnetic field data which we average over the spin period of about 3 s.

### 2.2    Data processing

Since the MP behaviour depends on the instantaneous properties of the solar wind, measurements of the same are required. Such measurements are provided by NASA/GSFC's OMNI data set through OMNIWeb of which we extracted 1-minute solar wind magnetic field and plasma data for the desired time range. The magnetic field information as well as the solar wind velocity is provided in an aberration corrected GSE coordinate system. Contrary, the extracted position information for a MP crossing underlies SW aberration. To calculate the individual aberration angle for each crossing, the SW velocity $v_{\mathrm{SW}}(t_0)$ within a time range of a few hours before the crossing is extracted from OMNI. From this the time $t_{\mathrm{SW}}(v_{\mathrm{SW}}(t_0))$ can be calculated which the solar wind needs to propagate from the Bow Shock Nose (BSN) to the MP position along the $x$-axis . The actual SW properties can then be extracted when the condition

$$\min\left(\left|t_{\mathrm{SW}}\left(v_{\mathrm{SW}}\left(t_0\right)\right) - t_0\right|\right) \tag{1}$$

is fulfilled. Afterwards position data and magnetic field data of each crossing can be corrected by the aberration angle and subsequently transformed into GSM coordinates. However, as OMNI data is prepared for the situation before the BSN, the transitional conditions at the BSN have to be taken into account. To do so, Rankine-Hugoniot conditions are applied to the SW velocity by multiplying with a factor of $1/4$. OMNI SW data is also taken as the input parameter for the Shue model.

To determine the MP normal direction, minimum variance analysis (MVA) (e.g., Paschmann and Daly, 1998) is applied to the magnetic field data within five minutes before and after any identified MP crossing, which will be defined below. As the MVA analysis only provides the orientation but not the direction of the normal, we assume the magnetopause normal to be always directed outwards of the MP, into the direction of the magnetosheath.

## 2.3   Identifying MP crossings

Time periods of possible MP crossings are manually selected from the available ESA and FGM data sets when the spacecraft is located near the MP position as predicted by the Shue model. Here, "near" means about $\pm 10 \, \mathrm{R_E}$ on the $xy$-plane around the predicted position. The actual crossings are subsequently identified by visual inspection of ESA and FGM measurements. The magnetosheath plasma is characterized by a significant energy flux around $1 \, \mathrm{keV}$. This flux almost instantly ceases once the MP has been crossed (Paschmann et al., 1993), see Fig. 1. Furthermore, also the particle number density, as derived from the energy spectrum, exhibits discontinuous changes when crossing the MP. In this way the precise crossing times and conditions are determined. Usually multiple crossings of the MP are also detected during the spacecraft motions into and out of the magnetotail. Like in Shue et al. (1997) the innermost crossing is selected for further analysis in the current study. In order to extend the analysis further, the outermost crossing is also considered separately. As the innermost crossing we denote the last (first) MP crossing of an outbound (inbound) pass through the boundary region. In the case of the outermost, it is exactly the opposite.

A total of 227 transitions is found this way, innermost and outermost crossings respectively. For 225 of these SW data is available. Figure 2 displays the spatial distribution of the MP positions determined on an $xd$-plane. $X$ is the GSM/GSE axis, whereas $d = \pm\sqrt{y^2 + z^2}$. The sign is equal to the sign of the $y$-component so that in- and outbound can be distinguished and either position can be visualized better. Using $d$ as a measure for the distance of the MP crossing from the $x$-axis supports the view of the model as axial symmetric and removes any projection errors in case of a projection onto any of the GSM planes. Shown as a red dot are the mean positions of each independent point clouds.

## 3   Comparison of position predictions

Figure 3 shows all necessary variables of the the Shue model and our convention to compare actual positions with it. As the Shue model fits empirical data, fitting parameters $a_1$ to $a_7$ for the standoff distance $r_0$ and the flaring parameter $\alpha$ come with

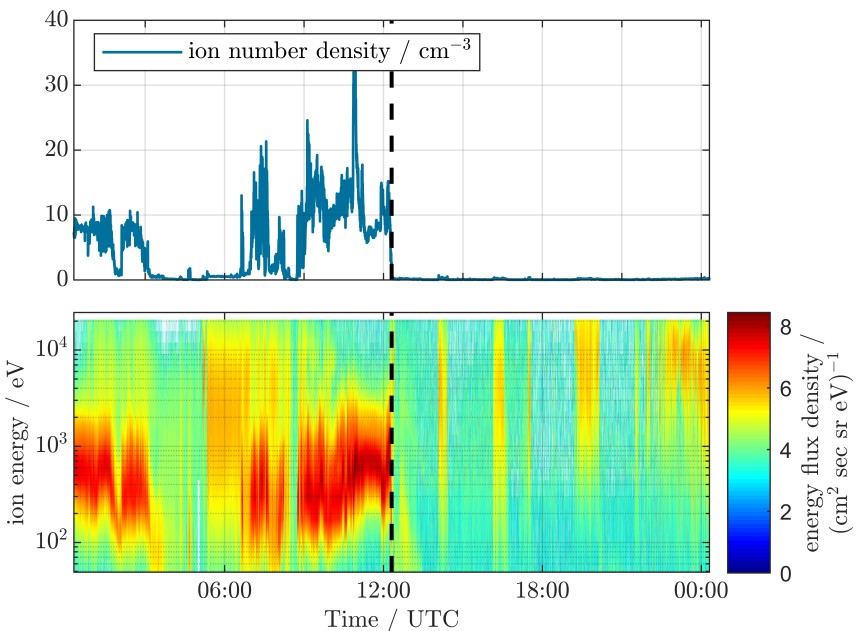

**Figure 1.** Example for MP crossing of THB on 24 April 2013. The probe comes from the magnetosphere and enters the magnetosheath at around 1218 UTC.

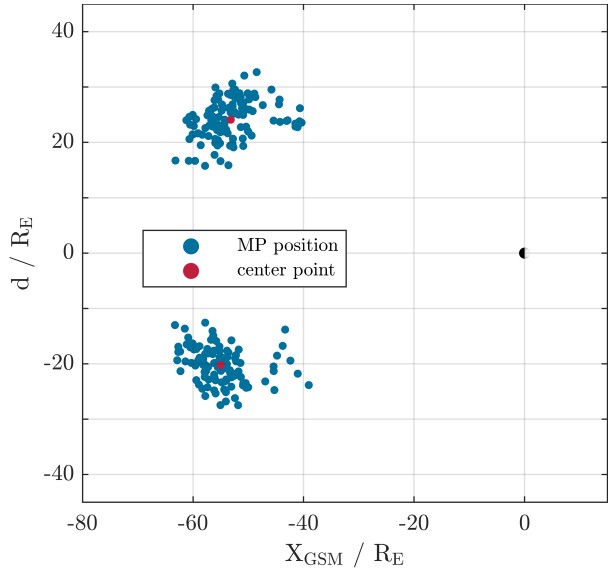

**Figure 2.** Distribution of MPs displayed on $x_{GSM}d$-plane, with $d = \pm\sqrt{y^2 + z^2}$, see text. The centre points of each independent point cloud is indicated by the red dots.

uncertainties (Shue et al., 1997).

$$r_0 = \begin{cases} (a_1 + a_2 B_z)(D_p)^{-\frac{1}{a_4}}, & \text{for } B_z \geq 0 \\ (a_1 + a_3 B_z)(D_p)^{-\frac{1}{a_4}}, & \text{for } B_z < 0 \end{cases} \tag{2}$$

$$\alpha = (a_5 + a_6 B_z)(1 + a_7 D_p) \tag{3}$$

Equations 2 and 3 are equations 10 and 11 in Shue et al. (1997).

5 We interpret this uncertainty as a measure of the standard deviation of the predicted MP position. If any MP position derived from ARTEMIS observations falls into this standard deviation, we regard this MP position as compatible with the model. This leads to a minimum and maximum modelled MP location, depending if the minimal or maximal error is added to the fitting parameters. These are shown as dashed lines in Fig. 3. Using the best fit parameters without any fitting errors lead to the mean MP location (solid line).

10 As the MP is almost parallel to the $x$-axis at lunar distances, we concentrate on differences between position prediction and actual position along the previously described $d$-axis, or, as we rotated all positions into the equatorial ($xy$-)plane, along the $y$-axis. Rotating into the equatorial plane or using the defined $d$-axis is equal to each other.

The standard deviation, which is the distance between minimum and maximum location along the $y$-axis, is called error range $\delta y$ by us, see Fig. 3. To quantify the actual MP position in relation to the model, its distance $\Delta y$ to the mean model

15 location is normalized to $\delta y/2$. We call this the normalized error of MP distance. Using this definition a MP laying exactly at the position predicted by the model has a normalized distance $2\Delta y/\delta y$ of zero. A MP laying exactly at the model MP with error has the normalized distance $2\Delta y/\delta y$ of $\pm 1$.

Figure 4 shows the distribution of normalized positions for the innermost crossing and Fig. 5 for the outermost crossing. In case of the innermost crossing the mean distance is at $-1.12$ with a standard deviation of $1.94$ and a skewness of $0.76$. This

20 means that the MP is usually found more close to the magnetotail as predicted by the model. In about $54\,\%$ of the crossings the model overestimates the location of the MP.

The situation is different with an outermost crossing, see Fig. 5. Here the mean distance is $2.48$ with a standard deviation of $2.37$ and a skewness of $0.89$. Accordingly the location of the MP is underestimated by the model.

The normalized MP distance does not show any strong correlation to the MP position along the $x_{\mathrm{GSM}}$-axis, the strength of

25 the SW $B_z$-component, or the SW speed. Each of the respective correlation coefficient is below $0.6$. As an example, Figures 6 and 7 display the scattering of the $x$-position against the normalized distance. Because of that, we conclude that there is no systematic deviation between modelled and actually observed MP distance with respect to these parameters.

## 4 Comparison of direction predictions

Besides its radial distance, the direction of the magnetopause normal can also be deduced from the Shue model and compared

30 with the observations at lunar distances. For this purpose model and observed normal directions are projected onto the $yz$-planes (polar plane) and $xy$-planes (equatorial), respectively, afterwards the deviation angles $\alpha$, respectively $\beta$, between model

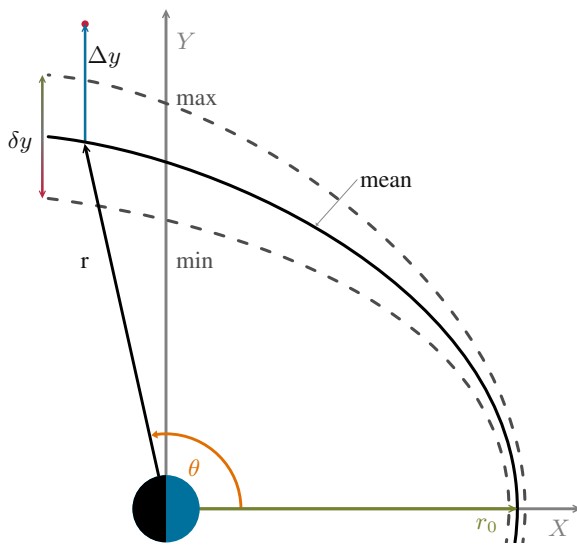

**Figure 3.** Scheme of the Shue et al. model and the normalization we use. The crossing of the MP with the $x$-axis is the standoff distance $r_0$. At certain angle $\theta$ the radial distance $r$ of the MP is given. Fitting uncertainties by the model are indicated by the dashed lines. Not shown is the flaring effect $\alpha$. To characterize the MP position (red dot) the difference distance along the $y$-axis between model and data, $\Delta y$, is normalized by half the error range, $\delta y$. (Graphic is not to scale.)

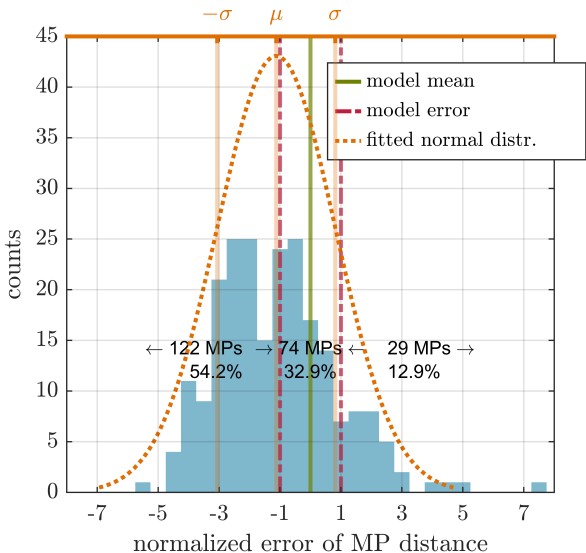

**Figure 4.** Normalized error of MP distance for innermost crossings. The error range is indicated by the vertical dashed dotted red line. $32.9\%$ of MP transitions lay within the model error. The mean is at $-1.12$, the standard deviation is $1.94$ and the skewness is $0.76$.

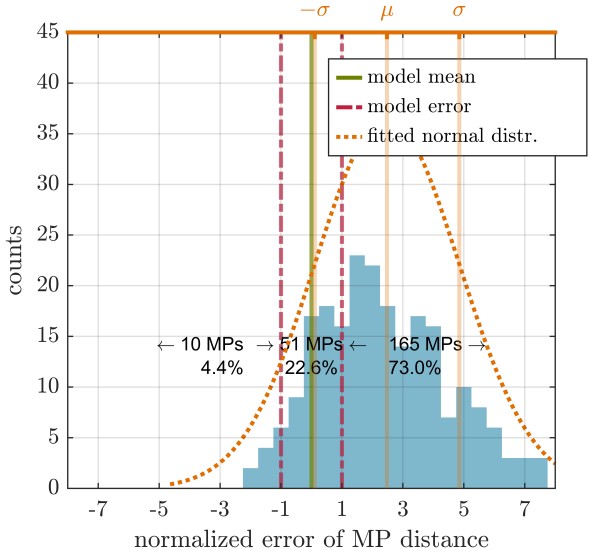

**Figure 5.** Normalized error of MP distance for outermost crossings. The error range is indicated by the vertical dashed dotted red line. 22.6 % of MP transitions lay within the model error. The mean is at 2.48, the standard deviation is 2.37 and the skewness is 0.89.

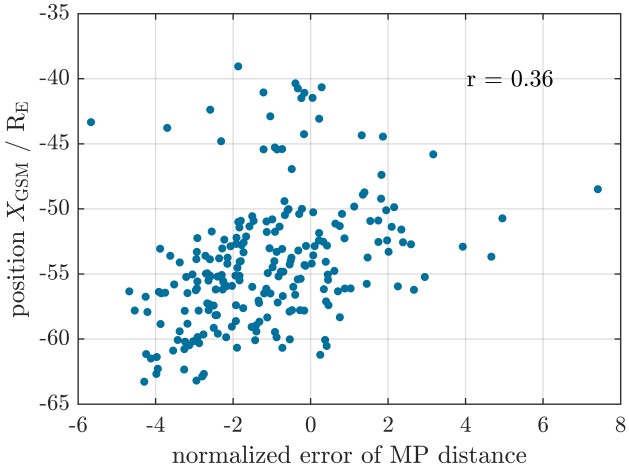

**Figure 6.** Example for non-existent correlation between the normalized MP distance and the position of the MP crossing projected onto the $x_{\mathrm{GSM}}$-axis for innermost crossings. The correlation coefficient is only $r = 0.36$.

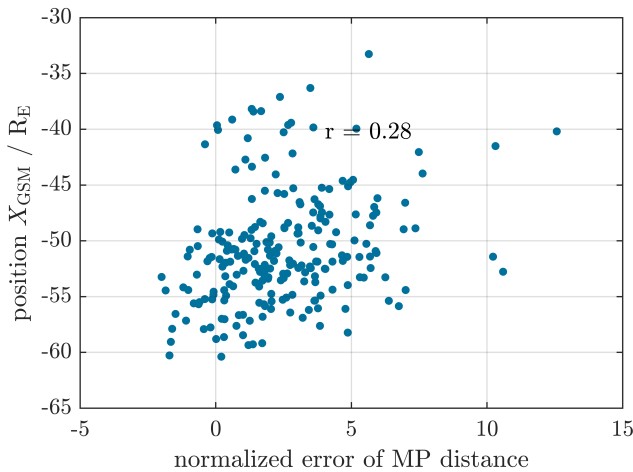

**Figure 7.** Example for non-existent correlation between the normalized MP distance and the position of the MP crossing projected onto the $x_{\mathrm{GSM}}$-axis for outermost crossings. The correlation coefficient is only $r = 0.28$.

and observed normal directions are determined. For deviation angles in the $yz$-plane ($xy$-plane) the sign of the angle is defined positive for situations in which the actual direction is pointing towards the positive $z$ ($x$) direction in relation to the model direction. Figure 8 illustrates this angle convention.

The thus defined deviation angles allow to highlight deviations of the magnetopause's opening angle, in case of the angle laying in $xy$-plane, which corresponds to the Shue flaring parameter, as well as deviations from the ideal axial symmetry, in case of the angle laying in the $yz$-plane. For each identified crossing solar wind data is used to calculate the model magnetopause. The expected distribution of angles $\gamma$ between the model normal direction and the $y_{\mathrm{GSM}}$-axis is shown in Fig. 9 for the innermost and Fig. 10 for the outermost crossings. Expected are angles with a mean of $4.6^\circ$ $(5.1^\circ)$ for the innermost (outermost) crossing directed sunwards, or positive direction, following our convention. This reinforces the assumption of a MP almost parallel to the $x$-axis, see Section 3.

Figures 11 and 12 display the deviation angle distributions.

For the innermost crossings we get the following results. The median deviation angles $\alpha$ for the $yz_{\mathrm{GSM}}$-plane are $1.9\,(44.6)^\circ$ for inbound crossings and $-7.1\,(45.9)^\circ$ for outbound. Values in parenthesis denote the respective standard deviation. For the angles $\beta$, the $xy_{\mathrm{GSM}}$-plane values are $8.0\,(38.3)^\circ$ for inbound and $5.0\,(42.7)^\circ$ for outbound. And results for the outermost crossings are as follows. The median deviation angles $\alpha$ for the $yz_{\mathrm{GSM}}$-plane are $-3.3\,(37.2)^\circ$ for inbound crossings and $6.1\,(42.7)^\circ$ for outbound. For the angles $\beta$, the $xy_{\mathrm{GSM}}$-plane values are $9.1\,(33.9)^\circ$ for inbound and $8.3\,(40.2)^\circ$ for outbound.

Both angles $\alpha$ and $\beta$ show median values near zero for all cases but come along with high scattering of more than $30^\circ$. Since we only observe one single crossing event per spacecraft and month, due to the spacecraft orbit, the high scattering is not surprising. But, with some caution, we conclude, that the predicted directions agree well the actual directions.

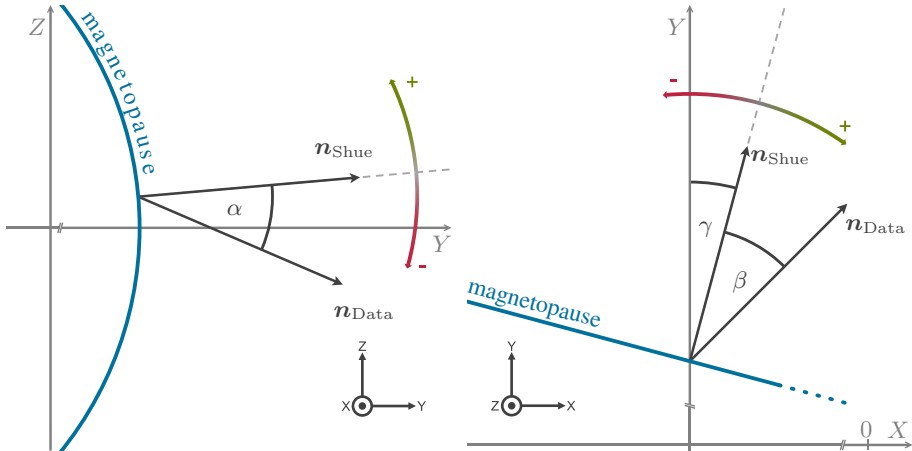

**Figure 8.** To compare the normal directions of the model and data, angles are measured as deviation from model direction. The angle $\alpha$ (front view, left panel) corresponds to deviation in the rotational symmetry, whereas $\beta$ (top view, right panel) corresponds to the MP flaring or short term perturbations. The expected opening angle $\gamma$ of the model is shown in Figures 9 and 10.

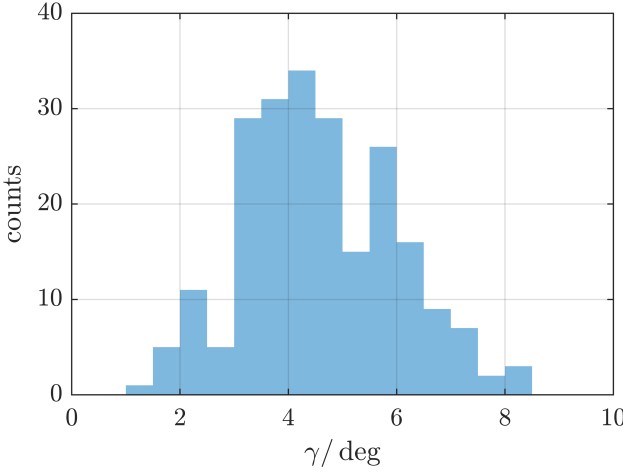

**Figure 9.** Angle between model MP normal direction of every crossing event and the $yz_{\mathrm{GSM}}$-plane, see angle $\gamma$ in Fig. 8. Expected are angles with a mean of $4.6°$.

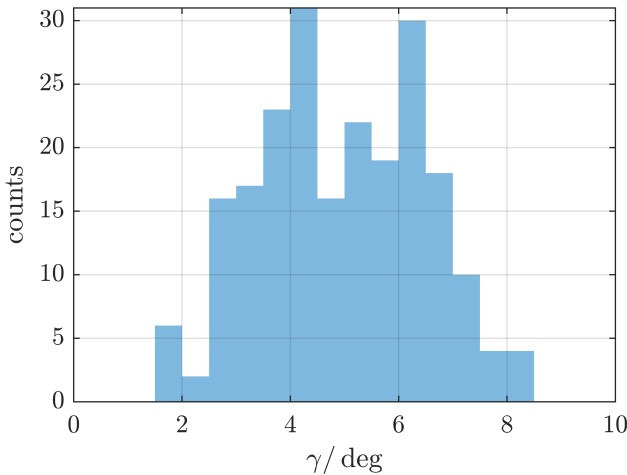

**Figure 10.** Angle between model MP normal direction of every crossing event and the $yz_{\mathrm{GSM}}$-plane, see angle $\gamma$ in Fig. 8. Expected are angles with a mean of $5.1°$.

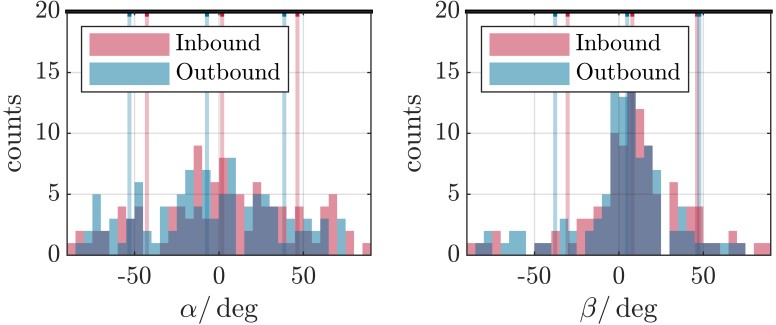

**Figure 11.** Deviation angle between model and data normal direction as projected onto $yz_{\mathrm{GSM}}$-plane (polar plane), left panel, and as projected onto $xy_{\mathrm{GSM}}$-plane (equatorial plane), right panel, for the innermost crossings. The distributions are separated by inbound (red) and outbound (blue) passes. Indicated by the coloured vertical lines are the respective median angles as well as the standard deviations, see text. The meaning of the angle sign is explained in Fig. 8.

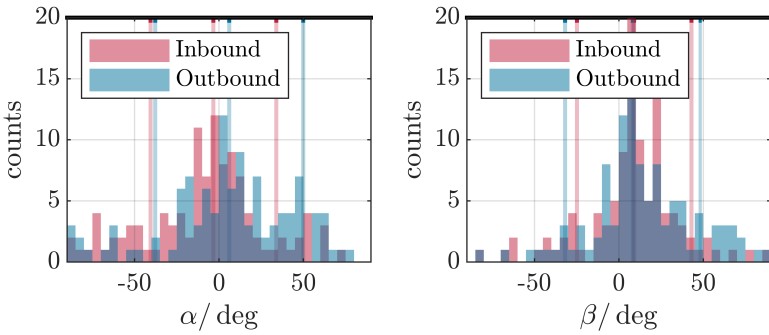

**Figure 12.** Deviation angle between model and data normal direction as projected onto $yz_{\text{GSM}}$-plane (polar plane), left panel, and as projected onto $xy_{\text{GSM}}$-plane (equatorial plane), right panel, for the outermost crossings. The distributions are separated by inbound (red) and outbound (blue) passes. Indicated by the coloured vertical lines are the respective median angles as well as the standard deviations, see text. The meaning of the angle sign is explained in Fig. 8.

## 5 Conclusions

The location of the magnetopause at lunar distances shows systematic differences to the model prediction. When choosing the innermost crossing of the MP, which is the same methodology as in Shue et al. (1997), the location is overestimated. In that case the MP is on average found much closer to the centre of the magnetotail. On the other hand, when choosing the outermost
crossing, Shue et al. underestimates the location and the MP is found in much greater distance to the magnetotail centre than expected.

Different to this are predictions about the normal direction of the MP. These scatter over a wider range of angles, but show a clear tendency to conform to the model predicted directions. Since the standard deviation is very large, it is not possible to make a well-founded statement about differences in in- and outbound traversals. Due to the high variability of the magnetopause
location caused by constantly changing solar wind conditions, the scattering in normal direction is as expected, since the SW induces directly changes in the MP standoff distance and indirectly surface waves such as Kelvin-Helmholtz instabilities due to differences in the plasma flow velocity. Essentially, the axial symmetry of the model can be confirmed for lunar distances in the magnetotail and near to the equatorial plane.

We conclude that the uncertainty in determination of the MP location increases with greater distance to the Earth. This
implies that the statistical width f the MP is larger than closer to Earth.

*Data availability.* THEMIS data and the latest calibration files are publicly available at http://themis.ssl.berkeley.edu/ or via the SPEDAS software.

*Competing interests.* The authors declare that they have no conflict of interest.

*Acknowledgements.* We acknowledge use of NASA/GSFC's Space Physics Data Facility's OMNIWeb service, and OMNI data. We acknowledge NASA contract NAS5-02099 and V. Angelopoulos for use of data from the THEMIS Mission - specifically, C. W. Carlson and J. P. McFadden for use of ESA data. This study is financially supported by the German Ministerium für Wirtschaft und Energie and the Deutsches Zentrum für Luft- und Raumfahrt under contract 50 OC 1403.

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
