# Peer review of "Statistical Analysis of Magnetopause Crossings at Lunar Distances"

_Annales Geophysicae, 2018_

## Referee Comment (RC1) · Anonymous Referee #1 · 25 Jul 2018

In this paper, the accuracy of the Shue et al. (1997) magnetopause model is evaluated at lunar distances from the Earth's center, by comparing model predictions to observations of the magnetopause by the ARTEMIS spacecraft. Lunar distances are significantly larger than the distances of the (observed) magnetopause locations from Earth that the model was originally based on. The authors find model predictions and observations of the magnetopause location to be in good agreement even at these large distances from Earth.

General comments ================

The analysis supports the conclusion that the model represents well the average location of the magnetopause at lunar distances, and this result merits publication. However, in my opinion, there are quite a number of issues in detail that need to be ad-

dressed before the paper is published: Other conclusions than the one stated above are not supported by results of the analysis. Some nomenclature is used inconsistently in the manuscript, some quantities are referred to by different terms and some terms/quantities are not thoroughly introduced and explained. The methodology differs in detail from Shue et al. (1997), the model to be evaluated. And the results of the MVA yielding local magnetopause normal directios are basically useless with respect to the evaluation of the model accuracy.

Specific comments ==================

Major issues ————

* page 2, line 21: The authors use a fixed value for the solar wind velocity. I think it would be very beneficial to use solar wind measurements from OMNI to determine the aberration angle specifically for each magnetopause crossing. A fixed aberration angle results in an additional source of variation in the observed magnetopause location (in the aberrated/model system); it may well account for that variation being larger than what is predicted by model errors (e.g. Figure 4).

* page 3, line 8 and Figure 1: Shue et al. specifically use the innermost crossings of the magnetopause. To validate their magnetopause model, the authors of this manuscript should use the same methodology. Otherwise the results will not necessarily be quantitatively comparable. The choice of the outermost crossings may be the reason for the "tendency of the magnetopause to be found at greater distance to the magnetotail than expected" (caption of Figure 4). Furthermore, Figure 1 seems to reveal that the authors do actually use the innermost magnetopause crossings, contrary to what is stated in the text. A zoom-in to the interval between 6 - 7 UT shows that THC crossed the magnetopause multiple times. I would identify the last (outermost) crossing at about 06:47:30 UT. The crossing at 06:14 UT indicated in the figure caption would be the innermost crossing, in my opinion.

* page 3, line 12: "projected onto the xy_GSM-plane": This is not good, because the

projection is not only used for illustrative purposes, but also to ascertain the accuracy of the model at lunar distances. The model is axis-symmetric around the x-axis of an aberrated GSE/GSM coordinate system. Hence, sqrt(yˆ2 + zˆ2) should be used as axis perpendicular to x for comparison with the model, and not a projection onto (aberrated) GSM xy. This issue affects all Figures and sections from 2.3 onwards, e.g.: usage of Delta y in section 3 and projections onto xy in section 4.

* page 4, line 17/18: A lack of correlation between normalized Delta y and x does not necessarily mean that there is no systematic deviation between model and actual magnetopause. It just means that the spread in Delta y is very large. If the model were perfect and the magnetopause not as dynamic as it is, we would expect Delta y to be zero over the entire range of x values. Hence, there would be a very high correlation.

* page 9, lines 5 to 10: There are a number of issues with the conclusions stated in this paragraph. The "tendency to agglomerate around the predicted directions" suggests that the flaring given by the model function is correct within the (very large) uncertainty limits of the angles determined by MVA (see large alpha and beta axis ranges in Figures 8 and 9). But this is to be expected should the model predict the magnetopause location accurately from the subsolar region to about 30 Earth radii downtail, as shown in Shue et al. (1997), and further on at 50-60 Earth radii downtail, as shown in this paper. Hence, in my opinion, the angular information inferred from the MVA is practically useless when evaluating the accuracy of the magnetopause model (including the assumption of axial symmetry). Again, the reason is the large scatter in the angles alpha and beta. The authors state that this scatter comes from the variability of the magnetopause position caused by constantly changing solar wind conditions, which I think is incorrect. Changing solar wind conditions should lead to a change in the magnetopause location, but not to a large change in normal directions. With MVA, the authors obtain estimates of instantaneous local normal directions, which will many times be very different to the reference or average normal direction even under constant solar wind conditions, due to the presence of surface waves or vortices (e.g.,

KHI).

Minor issues and technical corrections ————————————

\* page 1, line 2: "10 Earth radii": The way this sentence is written is confusing, because the magnetopause is usually more than 10 Earth radii away from the Earth's center, even in the subsolar region. The expression "10 Earth radii" probably refers only to the X-component of the locations.

\* page 1, line 3: "direction": At this point of the manuscript it is unclear that the authors refer to the direction normal to the magnetopause surface.

\* page 1, line 6: "reasonably": This could be quantified or described more accurately in the abstract.

\* page 1, line 9: "is defined": Pressure balance is a feature of the equilibrium magnetopause, not necesarily the definition of the boundary.

\* page 1, line 11: "very advanced": What does "very advanced" exactly mean here? I would rather say that the main virtue of the model is its simplicity.

\* page 1, line 11: "normal direction": Actually, the model only predicts the distance of the magnetopause as a function of angle to the Earth-Sun-line. Based on this function, reference/average normal directions may be determined for every point on the model magnetopause surface.

\* page 1, line 15: "found this form to be only depended": They only made it dependend on Bz and Dp, but did not necessarily test dependences on other solar wind parameters.

\* page 1, line 16/17: "axially symmetric around the x-axis in GSE and GSM": This is not correct. The model is valid in aberrated GSE or GSM coordinate systems, where the solar wind approaches Earth exactly along the x-axis (see beginning of section 2.2).

\* page 1, line 19: "detailed": What does this mean here?

* page 2, line 22: "unless otherwise indicated": Are there any indications? I have not found any.

* page 2, line 25: "five minutes before and after": How is this choice motivated? Would a different choice of intervals lead to better MVA eigenvalue ratios?

* page 2, line 31: "MP": Do the authors mean "spacecraft" here?

* page 3, line 2: The first sentence of section 2.3 sounds strange, because of the inserted subclause. Please reword.

* page 3, line 6: "changes crossing": Sounds strange.

* page 3, line 11: "is": Should be "are".

* page 3, line 14: "gather around their expected position": This sentence sounds some-what strange to me. In addition, I don't think that this can be seen in the figure, as each magnetopause crossing will take place under somewhat different solar wind conditions. Hence, the model magnetopause will look different each time. Figure 2, instead, only shows one model magnetopause for average conditions.

* page 4, first paragraph of section 3: The whole paragraph is written in a very con-fusing way. Please define and explain clearly in the text what is meant by: MP model range (use equations from Shue et al. if necessary), MP distance (I guess distance between the location of an ARTEMIS spacecraft at the time it was crossing the mag-netopause to the predicted model magnetopause along y), delta y, determined MP (I guess "location" is missing here), delta r / 2 (this is not even defined in Figure 3), and parallax errors (I am not sure this term is really applicable here, see also page 6 line 13).

* page 4, line 14: "very similar statistical properties": What does this mean? Please explain in more detail.

* page 4, line 16: Should be "coefficients".

[Figure]

* Figures 4 and 5: Define clearly "normalized error of MP distance" (used in both Figures 4 and 5), "relative MP position" as well as "normalized MP distance" in the captions. Why are three different terms used here?

* caption of Figure 5: "position of the MP projected": It is not possible to project the magnetopause surface onto a single point on the x axis. Please reword carefully.

* page 6, line 8: "opening angle": Maybe flaring angle?

* page 6, last paragraph: Use "°" instead of "degrees". Line 18: degrees is misspelled.

* page 6, last line: "one crossing per spacecraf and month": What is the reason for this restriction?

* page 7, second to last line: "the observed scattering is not surprising": I do not understand this. Even if there were more crossings in the data set, the scattering would not be any lower, I suppose. Please explain.

* page 9, lines 12/13: "most adequately" and "most important drivers": In reference to what? I am not sure the results of this study allow for any conclusion on the relative importance of Bz and Dp with respect to other parameters.
* * *

---

## Author Comment (AC1) · 16 Nov 2018

**Author Response to Reviewer comments**

**Major issues**

> *\* page 2, line 21: The authors use a fixed value for the solar wind velocity. I think it would be very beneficial to use solar wind measurements from OMNI to determine the aberration angle specifically for each magnetopause crossing. A fixed aberration angle results in an additional source of variation in the observed magnetopause location (in the aberrated/model system); it may well account for that variation being larger than what is predicted by model errors (e.g. Figure 4).*

We changed to use individual solar wind conditions to calculate the aberration angle for each crossing. To achieve this, OMNI measurements are propagated from the Bow Shock Nose to the MP location (on the x-axis).

> *\* page 3, line 8 and Figure 1: Shue et al. specifically use the innermost crossings of the magnetopause. To validate their magnetopause model, the authors of this manuscript should use the same methodology. Otherwise the results will not necessarily be quantitatively comparable. The choice of the outermost crossings may be the reason for the "tendency of the magnetopause to be found at greater distance to the magnetotail than expected" (caption of Figure 4). Furthermore, Figure 1 seems to reveal that the authors do actually use the innermost magnetopause crossings, contrary to what is stated in the text. A zoom-in to the interval between 6 - 7 UT shows that THC crossed the magnetopause multiple times. I would identify the last (outermost) crossing at about 06:47:30 UT. The crossing at 06:14 UT indicated in the figure caption would be the innermost crossing, in my opinion.*

All crossings were revaluated to get a complete list of all crossings, not just the outermost. This enabled us to choose the same methodology as Shue et al. and complement our analysis by choosing the innermost and outermost crossing respectively. We thank the author for this very helpful observation, because the accuracy of the investigation could be improved significantly.

> *\* page 3, line 12: "projected onto the xy_GSM-plane": This is not good, because the projection is not only used for illustrative purposes, but also to ascertain the accuracy of the model at lunar distances. The model is axis-symmetric around the x-axis of an aberrated GSE/GSM coordinate system. Hence, $sqrt(y^2 + z^2)$ should be used as axis perpendicular to x for comparison with the model, and not a projection onto (aberrated) GSM xy. This issue affects all Figures and sections from 2.3 onwards, e.g.: usage of Delta y in section 3 and projections onto xy in section 4.*

For better visualization we changed to the proposed representation.

> *\* page 4, line 17/18: A lack of correlation between normalized Delta y and x does not necessarily mean that there is no systematic deviation between model and actual magnetopause. It just means that the spread in Delta y is very large. If the model were perfect and the magnetopause not as dynamic as it is, we would expect Delta y to be zero over the entire range of x values. Hence, there would be a very high correlation.*

Unfortunately, we are not sure what is meant by this comment. Maybe it would be possible to clarify? From our point of view, the lack of correlation shows that the normalized error of MP distance does not have any systematic dependence on the named variables.

> *page 9, lines 5 to 10: There are a number of issues with the conclusions stated in this paragraph. The "tendency to agglomerate around the predicted directions" suggests that the flaring given by the model function is correct within the (very large) uncertainty limits of the angles determined by MVA (see large alpha and beta axis ranges in Figures 8 and 9). But this is to be expected should the model predict the magnetopause location accurately from the subsolar region to about 30 Earth radii downtail, as shown in Shue et al. (1997), and further on at 50-60 Earth radii downtail, as shown in this paper. Hence, in my opinion, the angular information inferred from the MVA is practically useless when evaluating the accuracy of the magnetopause model (including the assumption of axial symmetry). Again, the reason is the large scatter in the angles alpha and beta. The authors state that this scatter comes from the variability of the magnetopause position caused by constantly changing solar wind conditions, which I think is incorrect. Changing solar wind conditions should lead to a change in the magnetopause location, but not to a large change in normal directions. With MVA, the authors obtain estimates of instantaneous local normal directions, which will many times be very different to the reference or average normal direction even under constant solar wind conditions, due to the presence of surface waves or vortices (e.g., KHI).*

We hope the corresponding paragraph is now more clear, information on KHI is added.

"which will many times be very different to the reference or average normal": We think with looking to the angle distribution this problem is addressed. What we get is just an average situation from the angle distribution, which then agrees to the model deduced normal direction.

Maybe you could clarify your remark?

**Minor issues and technical corrections**

> *page 1, line 2: "10 Earth radii": The way this sentence is written is confusing, because the magnetopause is usually more than 10 Earth radii away from the Earth's center, even in the subsolar region. The expression "10 Earth radii" probably refers only to the X-component of the locations.*

Clarified that the x-axis is meant here.

> *page 1, line 3: "direction": At this point of the manuscript it is unclear that the authors refer to the direction normal to the magnetopause surface.*

"normal direction" is added.

> *page 1, line 6: "reasonably": This could be quantified or described more accurately in the abstract.*

Removed in the new version.

> *page 1, line 9: "is defined": Pressure balance is a feature of the equilibrium magnetopause, not necesarily the definition of the boundary.*

The definition of Baumjohan & Treuman, 1996, is added which defines the MP as the region between planetary magnetic field and solar wind plasma.

> *page 1, line 11: "very advanced": What does "very advanced" exactly mean here? I would rather say that the main virtue of the model is its simplicity.*

Changed to "simple" which is a more accurate description.

> *page 1, line 11: "normal direction": Actually, the model only predicts the distance of the magnetopause as a function of angle to the Earth-Sun-line. Based on this function, reference/average normal directions may be determined for every point on the model magnetopause surface.*

We added that the normal direction is only deduced from the model but is not originally predicted by it.

> *page 1, line 15: "found this form to be only depended": They only made it dependend on Bz and Dp, but did not necessarily test dependences on other solar wind parameters.*

Clarified that the model was designed in that way and not the result of an broader analysis.

> *page 1, line 16/17: "axially symmetric around the x-axis in GSE and GSM": This is not correct. The model is valid in aberrated GSE or GSM coordinate systems, where the solar wind approaches Earth exactly along the x-axis (see beginning of section 2.2).*

Aberration correction is added.

> *page 2, line 22: "unless otherwise indicated": Are there any indications? I have not found any.*

The subclause was removed.

> *page 2, line 25: "five minutes before and after": How is this choice motivated? Would a different choice of intervals lead to better MVA eigenvalue ratios?*

> *page 2, line 31: "MP": Do the authors mean "spacecraft" here?*

The MP is actually meant here, since the positions of the MP and the spacecraft coincide during the crossing.

> *page 3, line 2: The first sentence of section 2.3 sounds strange, because of the inserted subclause. Please reword.*

Sentence reworded: "Time periods of possible MP crossings are manually selected from the available ESA and FGM data sets when the spacecraft is located near the MP position as predicted by the Shue model. Here, "near" means about ± 10 $R_E$ on the xy-plane around the predicted position."

> *page 3, line 6: "changes crossing": Sounds strange.*

Reworded to "changes when crossing".

> *page 3, line 14: "gather around their expected position": This sentence sounds somewhat strange to me. In addition, I don't think that this can be seen in the figure, as each magnetopause crossing will take place under somewhat different*

*solar wind conditions. Hence, the model magnetopause will look different each time. Figure 2, instead, only shows one model magnetopause for average conditions.*

This is correct and the sentence is removed as well as the average Shue-MP from the Figure.

*\* page 4, first paragraph of section 3: The whole paragraph is written in a very confusing way. Please define and explain clearly in the text what is meant by: MP model range (use equations from Shue et al. if necessary), MP distance (I guess distance between the location of an ARTEMIS spacecraft at the time it was crossing the magnetopause to the predicted model magnetopause along y), delta y, determined MP (I guess "location" is missing here), delta r / 2 (this is not even defined in Figure 3), and parallax errors (I am not sure this term is really applicable here, see also page 6 line 13).*

The paragraph is rewritten in a hopefully more concise way.

*\* page 4, line 14: "very similar statistical properties": What does this mean? Please explain in more detail.*

With the new analysis this was removed.

*\* page 4, line 16: Should be "coefficients".*

With the new analysis this was removed.

*\* Figures 4 and 5: Define clearly "normalized error of MP distance" (used in both Figures 4 and 5), "relative MP position" as well as "normalized MP distance" in the captions. Why are three different terms used here?*

Also reworded and hopefully defined more concise.

*\* caption of Figure 5: "position of the MP projected": It is not possible to project the magnetopause surface onto a single point on the x axis. Please reword carefully.*

The location of the MP crossing was meant and the sentence was reworded.

*\* page 6, last paragraph: Use "°" instead of "degrees". Line 18: degrees is misspelled.*

Changed in the proposed way.

*\* page 6, last line: "one crossing per spacecraft and month": What is the reason for this restriction?*

This is due to the spacecraft orbit. We added that information.

*\* page 7, second to last line: "the observed scattering is not surprising": I do not understand this. Even if there were more crossings in the data set, the scattering would not be any lower, I suppose. Please explain.*

Unfortunately, we are not completely sure what was the problem with that.

*  page 9, lines 12/13: "most adequately" and "most important drivers": In reference to what? I am not sure the results of this study allow for any conclusion on the relative importance of Bz and Dp with respect to other parameters.*

Due to the new analysis this part was removed.

Overall, we would like to thank the reviewer for the good and helpful comments, which hopefully helped to improve our analysis.

[revised manuscript text omitted]